# Cancer-associated IDH mutations induce Glut1 expression and glucose metabolic disorders through a PI3K/Akt/mTORC1-Hif1α axis

Xun Liu[1], Kiyoshi Yamaguchi[1], Kiyoko Takane[1], Chi Zhu[1], Makoto Hirata[2,3], Yoko Hikiba[4], Shin Maeda[4], Yoichi Furukawa[1], Tsuneo Ikenoue[1]*

1 Division of Clinical Genome Research, Advanced Clinical Research Center, Institute of Medical Science, The University of Tokyo, Minato-ku, Tokyo, Japan, 2 Laboratory of Genome Technology, Institute of Medical Science, University of Tokyo, Minato-ku, Tokyo, Japan, 3 Department of Genetic Medicine and Services, National Cancer Center Hospital, Chuo-ku, Tokyo, Japan, 4 Department of Gastroenterology, Yokohama City University Graduate School of Medicine, Yokohama, Kanagawa Prefecture, Japan

* ikenoue@g.ecc.u-tokyo.ac.jp

**Data Availability Statement:** All relevant data are within the paper and its Supporting information files.

## Abstract

Isocitrate dehydrogenase 1 and 2 (*IDH1/2*) mutations and their key effector 2-hydroxyglutarate (2-HG) have been reported to promote oncogenesis in various human cancers. To elucidate molecular mechanism(s) associated with *IDH1/2* mutations, we established mouse embryonic fibroblasts (MEF) cells and human colorectal cancer cells stably expressing cancer-associated IDH1$^{R132C}$ or IDH2$^{R172S}$, and analyzed the change in metabolic characteristics of the these cells. We found that IDH1/2 mutants induced intracellular 2-HG accumulation and inhibited cell proliferation. Expression profile analysis by RNA-seq unveiled that glucose transporter 1 (Glut1) was induced by the IDH1/2 mutants or treatment with 2-HG in the MEF cells. Consistently, glucose uptake and lactate production were increased by the mutants, suggesting the deregulation of glucose metabolism. Furthermore, PI3K/Akt/mTOR pathway and Hif1α expression were involved in the up-regulation of Glut1. Together, these results suggest that Glut1 is a potential target regulated by cancer-associated *IDH1/2* mutations.

## Introduction

IDH1 and IDH2 are metabolic enzymes that catalyze oxidative decarboxylation of isocitrate to α-ketoglutarate (α-KG) [1–3]. Both IDH1 and IDH2 have hydrogenase activities, and generate NADPH using nicotinamide adenine dinucleotide phosphate (NADP$^+$) as electron acceptors [2–5]. Compared with IDH1 which is located in cytosol and peroxisomes, IDH2 is located in mitochondria [2–5], suggesting that these enzymes play a crucial role in metabolite exchange and electron transport in the cytosol and mitochondria.

Recurrent mutations of *IDH* were initially identified in gliomas by a cancer genome sequencing project [6]. Additional studies have revealed frequent *IDH1/2* mutations in a

**Funding:** The authors received no specific funding for this work.

**Competing interests:** The authors have declared that no competing interests exist.

variety of human cancers, including 70%-80% of low-grade gliomas, 50%-70% of chondrosarcomas, 10–20% of intrahepatic cholangiocarcinoma and approximately 20% of acute myeloid leukemia (AML) [3, 7, 8]. It is of note that *IDH1/2* mutations are usually heterozygous missense mutations, and that the mutations are primarily located at catalytic residues Arginine 132 (R132) of *IDH1* and Arginine 140 (R140) or Arginine 172 (R172) of *IDH2* [3]. These mutant proteins confer a neomorphic enzymatic activity resulting in the conversion from α-KG to 2-hydroxyglutarate (2-HG) [1, 9], and the accumulated 2-HG causes extensive anomalous effects on cell homeostasis. 2-HG blocks cell differentiation by competitive inhibition of αKG-dependent enzymes that are involved in epigenetic regulation [10], which induces additional alterations in cellular metabolism, redox state, and DNA repair [8, 11], suggesting that 2-HG functions as a potent "oncometabolite" [7].

Altered cellular energy metabolism is one of the "hallmarks of cancer", which are key biological characteristics acquired during carcinogenesis [12]. Two major biochemical events including increased glucose uptake and aerobic glycolysis are involved in the altered cellular energy metabolism [13]. GLUT1 encoded by the solute carrier family 2 member 1 (*SLC2A1*) gene plays a role in the uptake of glucose, and its expression is known to be regulated by hypoxia-inducible factor-1α (HIF1α) in hypoxemic conditions [14, 15]. The phosphatidylinositol 3-kinase (PI3K) /Akt signaling pathway is also reported to be associated with the regulation of GLUT1 and HIF1α expression [16, 17]. Enhanced GLUT1 expression during carcinogenesis has been identified in various malignancies, such as breast, lung, and pancreatic cancer (https://www.oncomine.org), which results in increased glucose uptake into cytoplasm of tumor cells [18].

Although the involvement of *IDH1/2* mutations in cancer has been reported, the precise mechanism(s) of mutant *IDH1/2* in carcinogenesis remains to be elucidated. In this study, we expressed cancer-associated hotspot IDH1/2 mutants in MEF cells or HCT116 cells and performed functional analysis. We identified *Slc2a1* as a novel downstream target of *IDH1/2* mutations, suggesting that GLUT1 may be useful as a biomarker of tumors harboring *IDH1/2* mutations.

## Materials and methods

### Cell culture

MEF cells were isolated from embryos of C57BL/6 mice at embryonic day 13.5 (ED13.5) and immortalized spontaneously by serial passages. HCT116, a human colorectal cancer cell line, was purchased from the American Type Culture Collection (Manassas, VAMEF). MEF cells and HCT116 cells were grown in Dulbecco's modified Eagle's medium (DMEM) (Thermo Fisher Scientific, Waltham, MA) and McCoy's 5A (modified) medium (Thermo Fisher Scientific), respectively, supplemented with 10% fetal bovine serum (FBS) (Thermo Fisher Scientific) and antibiotic/antimycotic solution (Sigma, St. Louis, MO). All animal experiments were approved by the Ethics Committee for Animal Experimentation and conducted in accordance with the Guidelines for the Care and Use of Laboratory Animals of the Institute of Medical Science and Department of Medicine, the University of Tokyo. Mice were euthanized with carbon dioxide followed by cervical dislocation.

### Reagents

Octyl-(R)-2HG, PI-103 and rapamycin were purchased from Sigma, Cayman (Ann Arbor, MI) and LC Laboratories (Woburn, MA), respectively.

## Retroviral plasmids and transduction

The wild-type (WT) cDNAs of human *IDH1* and *IDH2* were amplified by PCR using cDNA of SW480 cells as a template. The *IDH1^{R132C}* and *IDH2^{R172S}* mutant (MUT) cDNA were generated by a PCR-based site-directed mutagenesis and PCR amplification using cDNA of SW1353 cells carrying the mutation as a template, respectively. Both WT and MUT *IDH1/2* cDNAs were cloned into pCAGGSn3FC vectors to fuse a 3×FLAG tag at their C-terminus. The 3×FLAG tagged WT and MUT *IDH1/2* cDNAs were subcloned into a retroviral vector pMXs-puro (pMX-control) to obtain wild-type and mutant pMXs-3×FLAG-IDH1/2 (pMX-IDH1WT, pMX-IDH2WT, pMX-IDH1MUT and pMX-IDH2MUT) plasmids. Retroviral particles were produced by the transfection of PLAT-A packaging cells with pMX-IDH1WT, pMX-IDH2WT, pMX-IDH1MUT, pMX-IDH2MUT or pMX-control plasmids using Fugene 6 Transfection Reagent (Promega, Madison, WI). Viral supernatants were collected and used for infection. The cells expressing each gene were selected in the medium containing 2 μg/ml puromycin. After resistant cells to puromycin were selected, the cells stably expressing wild-type or mutant IDH1/2 (MEF-1WT, MEF-2WT, MEF-1MUT, and MEF-2MUT for MEF cells; HCT116-1WT, HCT116-2WT, HCT116-1MUT and HCT116-2MUT for HCT116 cells) and the cells transduced with control retrovirus (control MEF and control HCT116) were used for experiments.

## RNA-seq and gene set enrichment analysis

Total RNA was extracted from the MEF-2MUT cells, control MEF cells, and MEF cells treated with or without 2-HG using RNeasy Plus mini Kit (Qiagen, Valensia, CA). All experiments were carried out in triplicate. RNA integrity was evaluated using Agilent 2100 Bioanalyzer (Agilent Technologies, Santa Clara, CA), and RNA samples with RNA Integrity Number (RIN) > 8.8 were subjected to RNA-seq analysis. RNA-seq libraries were prepared with 100 ng of total RNA, using an Ion AmpliSeq Transcriptome Mouse Gene Expression kit (Thermo Fisher Scientific). The libraries were sequenced on the Ion Proton system using an Ion PI Hi-Q Sequencing 200 kit and Ion PI Chip v3 (Thermo Fisher Scientific), and the sequencing reads were aligned to AmpliSeq_Mouse_Transcriptome_V1_Reference using Torrent Mapping Alignment Program (TMAP). The data were analyzed using AmpliSeqRNA plug-in v5.2.0.3, a Torrent Suite Software v5.2.2 (Thermo Fisher Scientific), which provides QC metrics and normalized read counts per gene. Data processing was performed using the GeneSpring GX13.1 (Agilent Technologies). Genes with Benjamin-Hochberg-corrected p-values less than 0.05 were considered differentially expressed. Additionally, gene set enrichment analysis (GSEA) was performed using GSEA v4.1.0 for Windows with gene sets derived from hallmark collections, Pathway Interaction Database (PID) and BioCarta. KEGG pathway analysis was carried out using Molecular Signatures Database (MSigDB v7.2, http://www.broadinstitute.org/gsea/msigdb/index.jsp). All RNA-seq data were deposited to the NCBI sequence read archive (SRA) database under accession number GSE180369.

## Quantitative reverse-transcription PCR (qRT-PCR)

Total RNA was extracted from cultured cells using RNeasy Plus mini Kit (Qiagen). cDNA was synthesized from 1 μg of total RNA with Transcriptor First Strand cDNA Synthesis Kit (Roche Diagnostics GmbH, Mannheim, Germany). Quantitative PCR was performed using qPCR Kapa SYBR Fast ABI Prism Kit (Kapa Biosystems, Wilmington, MA) with sets of primers for *Ptgs2*, *Lamc2*, *Slc2a1* and *Hif1α* on StepOnePlus (Thermo Fisher Scientific). Sequences of the primers used are shown in Table 1. The levels of transcripts were determined by the relative

**Table 1. Sequence of primers used in qPCR.**

| Gene | Strand | Primer Sequence (5'>3') |
|---|---|---|
| Ptgs2 | Forward | GATGCTCTTCCGAGCTGTG |
| | Reverse | GGATTGGAACAGCAAGGATTT |
| Lamc2 | Forward | CTGGAGATCAGCAGCGAGA |
| | Reverse | TGCTGTCACATTAGCTTCCAA |
| Slc2a1 | Forward | TTACAGCGCGTCCGTTCT |
| | Reverse | TCCCACAGCCAACATGAG |
| Hif1α | Forward | CATGATGGCTCCCTTTTTCA |
| | Reverse | GTCACCTGGTTGCTGCAATA |
| Gapdh | Forward | TGTCCGTCGTGGATCTGAC |
| | Reverse | CCTGCTTCACCACCTTCTTG |

standard curve method, and glyceraldehyde-3-phosphate dehydrogenase (*Gapdh*) was used as an internal control.

## Western blotting

Total protein was extracted from cultured cells using radioimmunoprecipitation assay (RIPA) buffer (50 mM Tris-HCl, pH8.0, 150 mM sodium chloride, 0.5% sodium deoxycholate, 0.1% sodium dodecyl sulfate, 1.0% NP-40) supplemented with the Protease Inhibitor Cocktail Set III (Calbiochem, San Diego, CA) and a phosphatase inhibitor cocktail PhosSTOP™ (Roche). Protein concentration was determined by BCA Protein Assay Kit (Thermo Fisher Scientific). Protein (30–50 μg/lane) was separated by 10% SDS-PAGE and transferred on to a polyvinylidene fluoride (PVDF) membrane (GE Healthcare, Buckinghamshire, UK). After the blocking with 5% skim milk in TBS-T (Tris-buffered saline-Tween20) for 1 h, the membranes were incubated overnight with primary antibodies including anti-Flag (F3165, Sigma), anti-Glut1 (ab115730, Abcam, Cambridge, UK), anti-phospho-S6k (9205, Cell Signaling Technology, Danvers, MA), anti-total-S6k (2708, Cell Signaling Technology), anti-phospho-Akt (Ser473) (4060, Cell Signaling Technology), anti-phospho-Akt (Thr308) (13038, Cell Signaling Technology), anti-total-Akt (4681, Cell Signaling Technology), anti-Hif1α (14179, Cell Signaling Technology), anti-IDH1 (D2H1) (8137, Cell Signaling Technology), anti-IDH2 (D8E3B) (56439, Cell Signaling Technology), and anti-β-actin (A5441, Sigma). All antibodies except anti-Glut1 (1:50,000) were diluted by 1:1000. Horseradish peroxidase-conjugated goat anti-mouse or anti-rabbit IgG (GE Healthcare) served as the secondary antibody for the ECL Detection System (GE Healthcare).

## Cell proliferation assay

Cell proliferation assay was carried out by water soluble tetrazolium salts (WST)-based colorimetric method using Cell-counting kit-8 according to the manufacturer's recommendations (Dojindo, Kumamoto, Japan). Absorbance was measured at 450 nm using FLUOstar OPTIMA Microplate Reader (BMG Labtechnologies, GmbH, Germany).

## 2-HG measurement

An enzymatic 2-HG assay was used to determine the intracellular 2-HG concentration. Lysates of $5 \times 10^6$ cells were collected and their intracellular 2-HG content was measured using a D-2-HG Assay Kit (Abcam). Absorbance was measured by FLUOstar OPTIMA Microplate

Reader (BMG Labtechnologies) at 450nm. D-2-HG concentrations of indicated cell lysates were calculated according to manufacturer's instructions.

## Glucose uptake assay

Cells were plated in a 12-well cell culture plate and incubated overnight. Glucose uptake was measured by Glucose Uptake-Glo™ Assay (Promega), according to the manufacturer's protocol. Cell viability was quantified simultaneously by CellTiter-Glo® Luminescent Cell Viability Assay (Promega) for normalization.

## Lactate level assay

Lysates of $2\times10^4$ cells were collected, respectively. Intracellular lactate levels were then measured by Lactate-Glo™ Assay (Promega), according to manufacturer's protocol. Cell viability was quantified simultaneously by CellTiter-Glo® Luminescent Cell Viability Assay (Promega) for normalization.

## Gene silencing

Small interfering RNA (siRNA) targeting *Hif1α* (SASI_Mm01_00070473), *Akt1* (Mm_Akt1_3533), *Akt2* (Mm_Akt2_5904), and *Akt3* (Mm_Akt3_0790) were purchased from Sigma and control siRNA (ON-TARGETplus Non-targeting Pool #D-001810-10) was purchased from GE Dharmacon (Lafayette, CO). Target sequences of the siRNAs are shown in Table 2. Cells were seeded one day before the treatment with siRNA, and transfected with 10 nM of the aforementioned siRNAs using Lipofectamine RNAiMAX (Thermo Fisher Scientific). Forty-eight hours after the transfection, RNA and proteins were extracted from the cells. The silencing effect of siRNAs was evaluated by real-time qPCR and Western blotting.

## Statistical analysis

Statistical analysis was performed by Student's t-test with Benjamini-Hochberg correction for the analysis of gene expression profiles. Unpaired Student's t-test was used for the statistical analysis of cell proliferation, qRT-PCR, glucose uptake assay and lactate level assay.

# Results

## Analysis of 2-HG production and cellular proliferation in MEF and HCT116 cells expressing cancer-associated *IDH1/2* mutations

To investigate the function of IDH1/2 mutants (IDH1[R132C] and IDH2[R172S]) in carcinogenesis, MEF cells and HCT116 cells that stably express exogenous wild-type or mutant IDH1/2 (MEF-1WT, MEF-2WT, MEF-1MUT, and MEF-2MUT for MEF cells; HCT116-1WT, HCT116-2WT, HCT116-1MUT, and HCT116-2MUT for HCT116 cells) were established using a retrovirus transduction system (Fig 1A, S1 and S2A Figs). To evaluate the converting

**Table 2. Sequence of siRNA.**

| siRNA | Sequence |
|---|---|
| Control | ON-TARGETplus Non-targeting Control Pool #D-001810-10 |
| *Hif1α* | CAAGCAACUGUCAUAUAUA |
| *Akt1* | GUGAUUCUGGUGAAAGAGA |
| *Akt2* | GAGAUGUGGUGUACCGUGA |
| *Akt3* | CUGUUAUAGAGAGAACAUU |

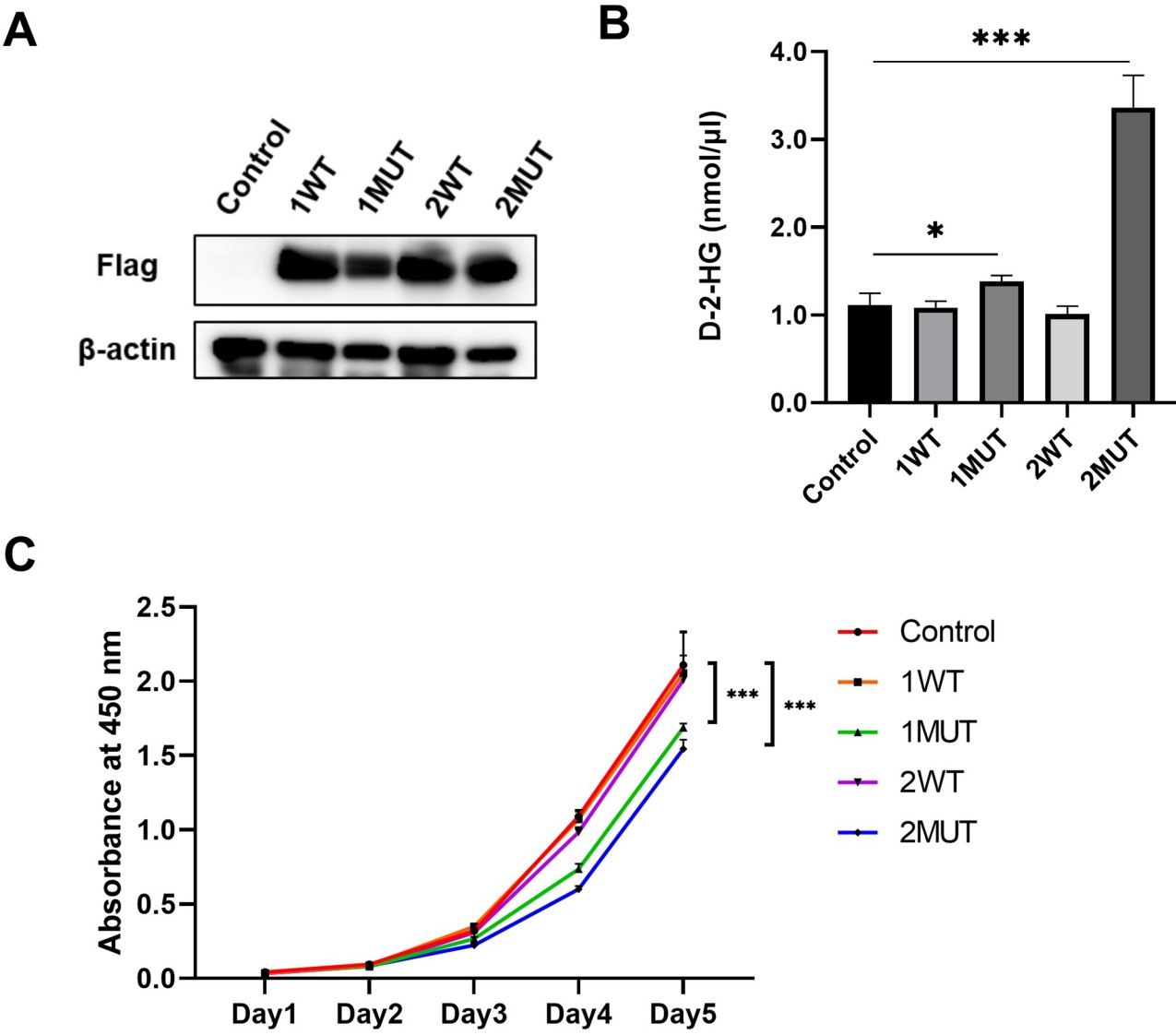

**Fig 1. Effect of wild type and mutant IDH1/2 on 2-HG and cellular proliferation.** (A) Western blot analysis of the MEF-1WT, MEF-1MUT, MEF-2WT, MEF-2MUT and control MEF cells. Expression of β-actin served as an internal control. (B) Concentration of 2-HG in the lysates from cells indicated in (A). The data represent mean ± SD of three independent experiments. * p<0.05, *** p<0.001. (C) The proliferation of cells was measured by WST-8 assay. The data represent mean ± SD of three independent experiments. *** p<0.001.

activity of the mutant IDH1/2 proteins from α-KG to 2-HG, we measured 2-HG accumulation in MEF cells and HCT116 cells using an enzymatic assay (Fig 1B and S2B Fig). Expression of the mutant IDH1 and IDH2 induced 2-HG accumulation in MEF cells (MEF-1MUT and MEF-2MUT cells) by 1.25-fold and 3.03-fold, respectively, compared to the control cells. Similarly, mutant IDH1 and IDH2 augmented 2-HG accumulation in HCT116 cells (HCT-1MUT and HCT-2MUT cells) by 1.84 and 5.16 fold, respectively, compared to HCT116 control cells. These data indicate that both mutants enhance the conversion from α-KG to 2-HG, and that the mutant IDH2 exhibits a stronger effect on the conversion to 2-HG compared with the mutant IDH1.

Next, we analyzed the effect of these mutants on cell proliferation. The proliferation of the MEF cells and HCT116 cells were examined using a WST-8 assay. Unexpectedly, exogenous

expression of the mutant IDH1 as well as the mutant IDH2 slightly suppressed the proliferation of MEF-1MUT and MEF-2MUT, compared to the control cells, MEF-1WT, or MEF-2WT (Fig 1C). Although exogenous expression of wild type IDH1 or IDH2 slightly suppressed the proliferation of HCT116 cells, that of mutant IDH1 or IDH2 showed greater suppression than the wild types (S2C Fig). In addition, mutant IDH2 was more effective than mutant IDH1 on the growth retardation.

## Identification of pathways and genes commonly regulated by cancer-associated IDH2 mutant and 2-HG

To clarify the effect of IDH2 mutant through accumulated 2-HG, we performed RNA-seq analysis. Comparison of expression profiles of the MEF cells expressing mutant IDH2 (2MUT) with control MEF cells (Control) identified a total of 575 up-regulated and 716 down-regulated genes (Fig 2A). We also compared expression profiles of MEF cells treated with 300 μM of membrane-permeant 2-HG with non-treated MEF cells, and found a total of 884 up-regulated genes and 1046 down-regulated genes by the 2-HG treatment. Combination of these data identified a total of 117 genes including 37 commonly up-regulated genes and 80 commonly down-regulated genes by the IDH2 mutant and 2-HG treatment (S1 Table).

We then carried out GSEA for the gene profiles altered by the mutant IDH2, and those altered by 2-HG (Fig 2B and 2C, S2 and S3 Tables). The analysis showed enrichment of signatures corresponding to "PI3K/Akt/mTOR signaling" and "glycolysis" in cells expressing the IDH2 mutant as well as in cells treated with 2-HG. In addition, gene sets correlated with "HIF1 pathway" were enriched in the *IDH2* mutant cells but not in the cells with 2-HG treatment. On the other hand, genes associated with "VEGF pathway" were enriched by 2-HG treatment, but not in the *IDH2* mutant cells.

Next, to understand the biological alterations by the *IDH2* mutation through 2-HG accumulation, KEGG pathway analysis with the 117 overlapped genes was performed using curated gene sets in the MSigDB. We found significant enrichment of genes associated with "ECM-receptor interaction", "focal adhesion", "TGF-beta signaling pathway", "pathways in cancer", "cytosolic DNA-sensing pathway", and "cell adhesion molecules" (Fig 2D, Table 3). These data suggest that accumulated 2-HG may alter communication between cells and matrix, cellular adhesion, and signal transduction pathways, and that these biological alterations may play a role in human carcinogenesis.

## Identification of Glut1 as a target molecule induced by the IDH1/2 mutants and 2-HG

To identify genes involved in *IDH1/2* mutation-associated carcinogenesis, we searched for genes commonly altered by the IDH2 mutant and the treatment with 2-HG. In the six enriched gene sets detected by KEGG pathway analysis, we focused on the six genes in "pathways in cancer" (Table 3). Three genes, *Ptgs2*, *Lamc2* and *Slc2a1*, were up-regulated, and three, *Bmp4*, *Mmp9* and *Jup*, were down-regulated (Table 3). Quantitative RT-PCR analysis confirmed that expression of *Ptgs2*, *Lamc2*, and *Slc2a1* were elevated by the IDH2 mutant (Fig 3A–3C) and 2-HG treatment (Fig 3D–3F). Additionally, the expression levels of these genes were similarly enhanced by the IDH1 mutant (Fig 3A–3C).

Since *Slc2a1* encodes Glut1, a key molecule involved in cellular energy metabolism, we next focused on this molecule. Western blot analysis further corroborated increased expression of Glut1 by the IDH mutants (Fig 3G) and the treatment with 2-HG in MEF cells (Fig 3H). Consistently, both mutant IDH1 and IDH2 also induced Glut1 expression significantly in HCT116 cells (S2A Fig).

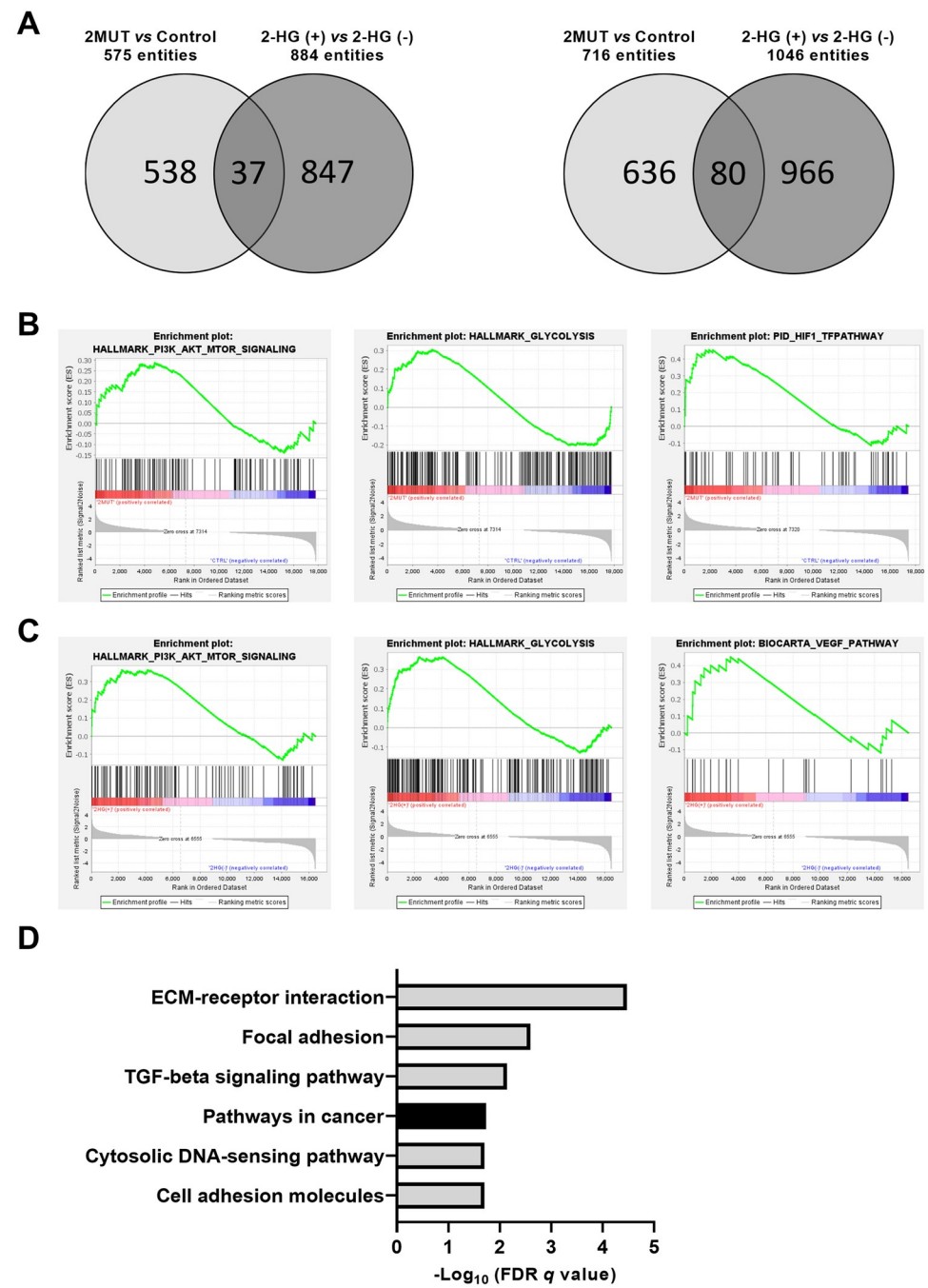

**Fig 2. Genes with altered expression by the IDH2 mutant or 2-HG treatment.** (A) Venn diagram of up-regulated (left) and down-regulated (right) genes by the overexpression of the IDH2 mutant or the treatment with 2-HG in MEF cells. (B) GSEA plot of enriched PI3K_AKT_MTOR signaling, glycolysis, and HIF1_TF pathway gene sets by the expression of the IDH2 mutant in MEF cells. (C) GSEA plots of enriched PI3K_AKT_MTOR signaling, glycolysis and VEGF pathway gene sets by the treatment with 2-HG in MEF cells. (D) FDRs of enriched gene sets by the IDH2 mutant and 2-HG treatment in KEGG pathway analysis.

## Enhanced glucose uptake and glycolysis by the IDH1/2 mutants

Since Glut1, a glucose transporter that facilitates glycolysis, was up-regulated by the IDH1/2 mutants at least in part through the increase of 2-HG, we investigated the glucose uptake in

**Table 3. Details of KEGG pathway analysis by MSigDB.**

| Name of Gene Set | Description | Overlapped gene symbols | p-value | FDR q-value |
|---|---|---|---|---|
| KEGG_ECM_RECEPTOR_INTERACTION | ECM-receptor interaction | *Lamc2, Thbs2, Col1a2, Col3a1, Col11a1, Sdc2* | 1.82E-07 | 3.38E-05 |
| KEGG_FOCAL_ADHESION | Focal adhesion | *Lamc2, Thbs2, Col1a2, Col3a1, Col11a1, Cav1* | 2.72E-05 | 2.53E-03 |
| KEGG_TGF_BETA_SIGNALING_PATHWAY | TGF-beta signaling pathway | *Thbs2, Bmp4, Dcn, Smad1* | 1.16E-04 | 7.17E-03 |
| KEGG_PATHWAYS_IN_CANCER | Pathways in cancer | *Lamc2, Bmp4, Slc2a1, Ptgs2, Mmp9, Jup* | 3.96E-04 | 1.84E-02 |
| KEGG_CYTOSOLIC_DNA_SENSING_PATHWAY | Cytosolic DNA-sensing pathway | *Ccl5, Sting1, Zbp1* | 5.67E-04 | 1.98E-02 |
| KEGG_CELL_ADHESION_MOLECULES_CAMS | Cell adhesion molecules (CAMs) | *Sdc2, Hla-a, Nlgn2, Pvr* | 6.39E-04 | 1.98E-02 |

MEF cells and HCT116 cells, using an enzymatic assay. As we expected, the capacity of glucose uptake in both MEF cells (MEF-1MUT and MEF-2MUT) and HCT116 cells (HCT116-1MUT and HCT116-2MUT) was significantly enhanced by the IDH1/2 mutants (Fig 4A and S2D Fig).

We further measured possible alteration of intracellular lactate, the end product of glycolysis. In concert with the elevated expression of Glut1, an increase of intracellular lactate level was observed in MEF-1MUT and MEF-2MUT, suggesting that the up-regulation of Glut1 by IDH1/2 mutants contributes to the induction of glycolysis in MEF cells (Fig 4B). On the other hand, mutant IDH1 and IDH2 did not exhibit significant increase of intracellular lactate level in HCT116 cells, indicating that augmented Glut1 expression induced by IDH1/2 mutants may not be sufficient to affect glycolytic pathway in colorectal cancer cells (S2E Fig).

## Involvement of PI3K/Akt/mTOR pathway in the regulation of Glut1 induction by cancer-associated *IDH1/2* mutations

It was reported that *IDH1/2* mutations activate the PI3K/Akt/mTOR pathway [19]. Consistently, the GSEA analyses identified the association of the *IDH2* mutation with PI3K/Akt/mTOR signaling (Fig 2B). Thus, we investigated phosphorylation of Akt and ribosomal protein S6 kinase (S6k) in MEF cells and HCT116 cells. As a result, expression of phosphorylated S6k and Akt on Ser473 were increased by mutant IDH1 as well as mutant IDH2 (Fig 5A and S2A Fig), indicating both mTORC1 and mTORC2 were activated. In addition, the IDH1/2 mutants also induced elevated phosphorylation of Akt on Thr308 (Fig 5A and S2A Fig), which

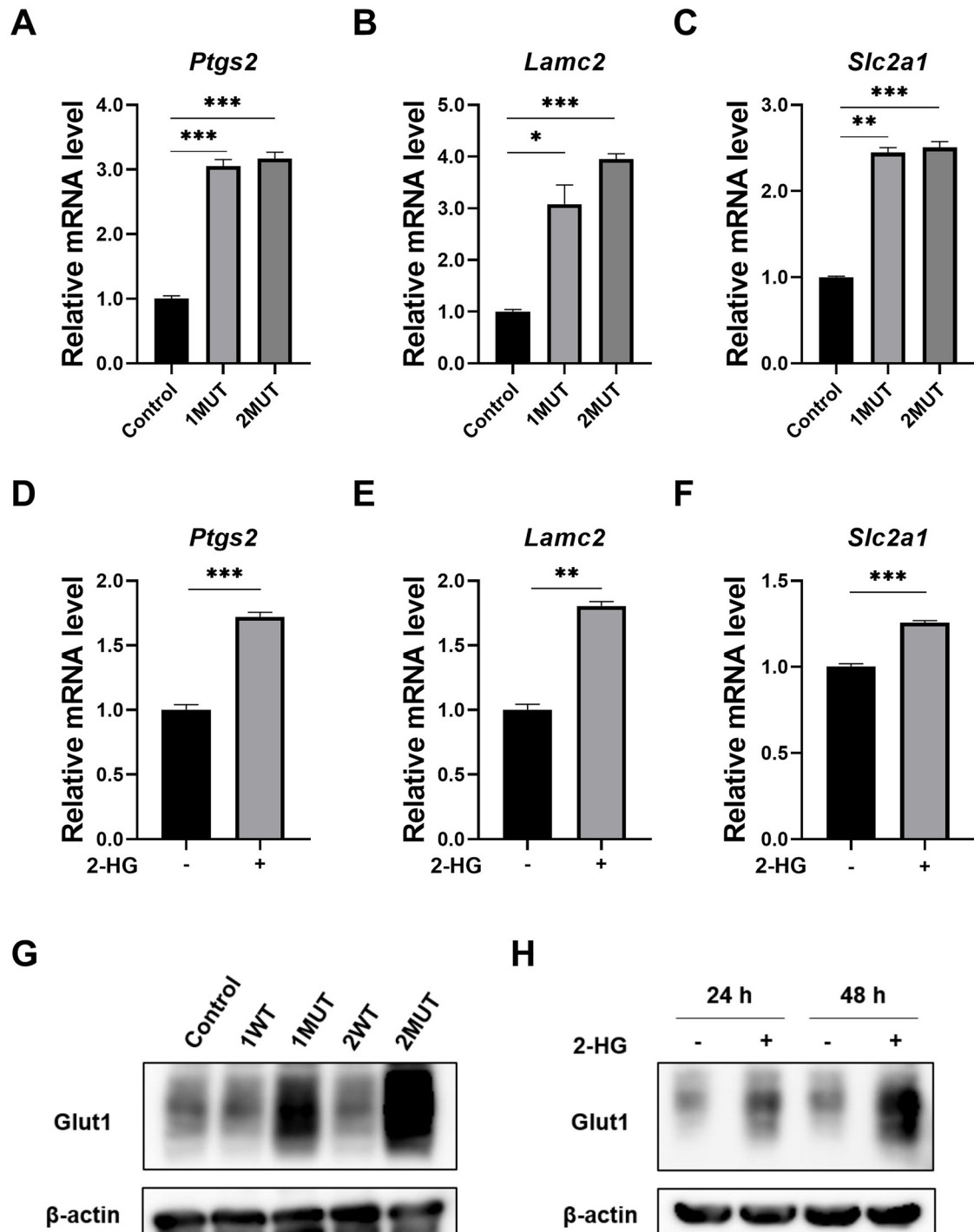

**Fig 3. Enhanced expression of Glut1 by the expression of IDH1/2 mutants and 2-HG.** (A–C) Induction of *Ptgs2* (A), *Lamc2* (B), and *Slc2a1* (C) in MEF cells stably expressing the IDH1 or IDH2 mutant. Expression was determined by real time-PCR. Expression of *Gapdh* was used as an internal control. The data represent mean ± SD of three independent experiments. * $p < 0.05$, ** $p < 0.01$, *** $p < 0.001$. (D–F) Induction of *Ptgs2* (D), *Lamc2* (E), and *Slc2a1* (F) in MEF cells treated with 300 μM of 2-HG for 24 h. ** $p < 0.01$, *** $p < 0.001$. (G) Elevated expression of Glut1 protein by the IDH1/2 mutants. Expression of β-actin served as an internal control. The data are representative of three independent experiments. (H) Elevated expression of Glut1 protein in response to the treatment with 300 μM of 2-HG. Cell lysates were harvested at 24 h or 48 h after treatment and subjected to Western blotting.

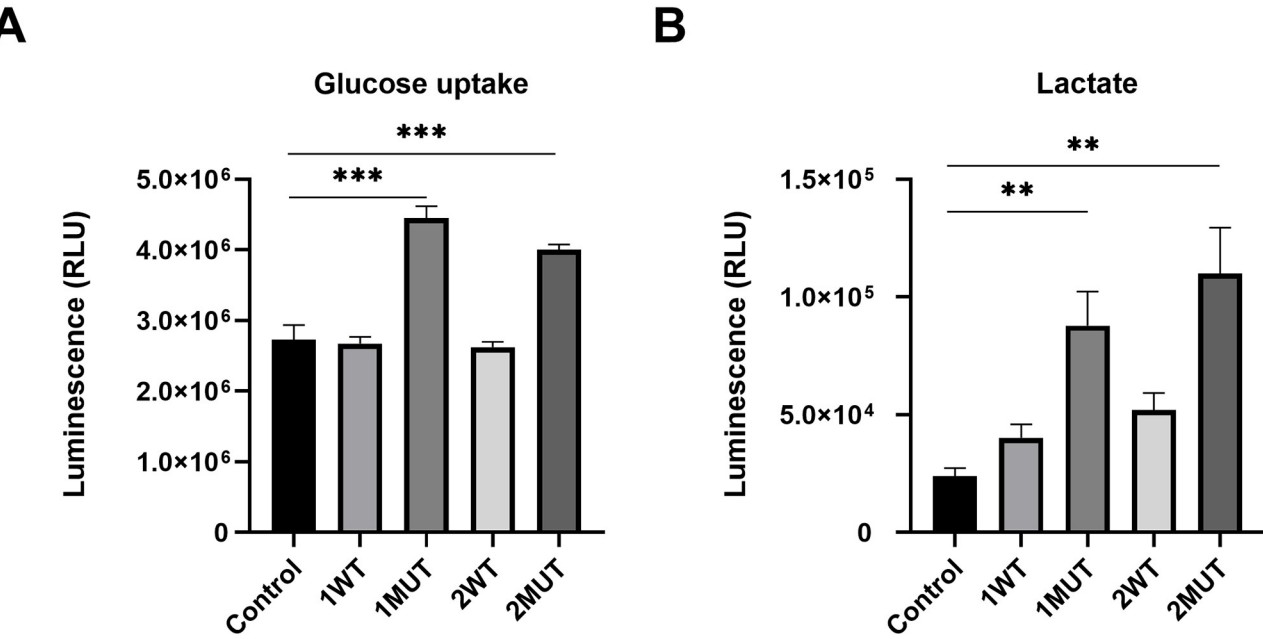

**Fig 4. Augmentation of glucose uptake and intracellular lactate levels by the IDH1/2 mutants.** (A) Glucose uptake levels in MEF-1WT, MEF-1MUT, MEF-2WT, MEF-2MUT, and the control cells were measured by a bioluminescent assay based on the detection of 2DG6P. The data represent mean ± SD of three independent experiments. *** $p < 0.001$. (B) Intracellular lactate levels in the MEF cells indicated in (A) were measured by a bioluminescent assay for the detection of L-lactate. The data represent mean ± SD of three independent experiments. ** $p < 0.01$.

is regulated by PDK1. These data indicate activation of the PI3K/PDK1/Akt/mTOR cascade by the *IDH1/2* mutations. We further studied whether the induced Glut1 expression by the *IDH1/2* mutations is regulated through the activation of PI3K/Akt/mTORC1 pathway in MEF cells. As shown in Fig 5B, knockdown of *Akt1*, *Akt2* and *Akt3* by siRNA decreased Glut1 expression in the MEF cells with the IDH1/2 mutants. In addition, treatment with PI-103, a multi-targeted PI3K inhibitor, and rapamycin, an inhibitor of mTORC1, markedly reduced Glut1 expression (Fig 5C and 5D), suggesting that PI3K/Akt/mTORC1 pathway is involved in the increased Glut1 expression by the IDH mutants.

### Involvement of Hif1α in the regulation of Glut1 by the *IDH1/2* mutations

Reportedly, GLUT1 expression is augmented under hypoxia through the induction of HIF1α in cancer cells [15]. The GSEA analysis in the current study also showed that genes correlated with the "HIF1 pathway" were enriched in MEF-2MUT (Fig 2B). Therefore, we investigated whether the IDH1/2 mutants enhance the expression of Hif1α. Western blot analysis showed that exogenous expression of the IDH1 or IDH2 mutant increased the expression levels of Hif1α in MEF cells and HCT116 cells (Fig 6A and S2A Fig). Furthermore, knockdown of Hif1α reduced Glut1 expression in MEF-1MUT and MEF-2MUT cells (Fig 6A). To investigate the association between PI3K/Akt/mTORC1 pathway and Hif1Iα expression in *IDH1/2* mutant cells, we treated MEF-1MUT, MEF-2MUT, and control cells with rapamycin. As a result, qPCR analysis revealed that treatment of rapamycin reduced Hif1α expression by 25%, 36%, and 30%, in control, MEF-1MUT, and MEF-2MUT cells, respectively (Fig 6B). It is of note that rapamycin markedly decreased the Hif1α protein in these cells (Fig 6C), suggesting that both transcriptional and post-transcriptional regulation play a role in the expression of Hif1α in the

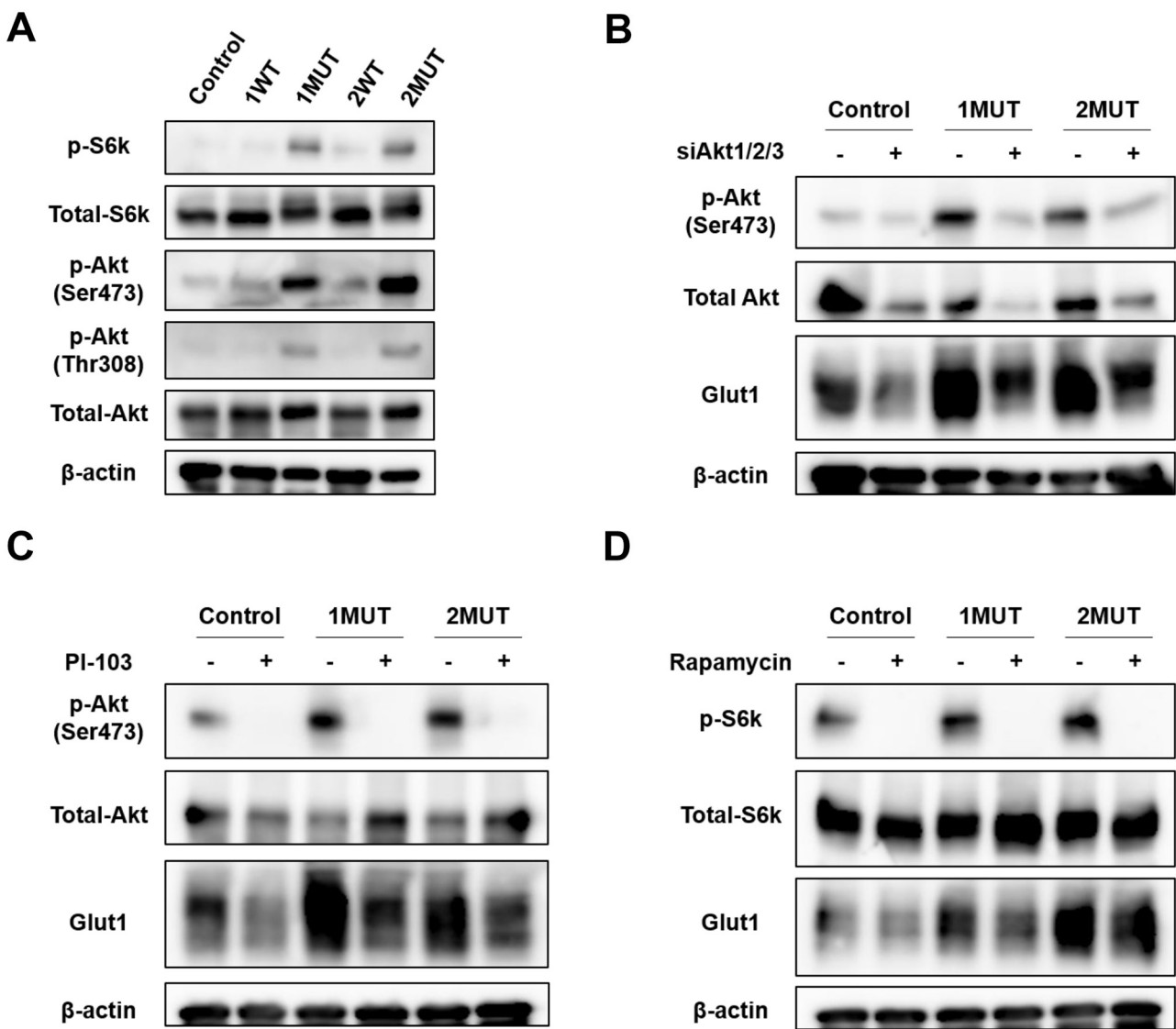

**Fig 5. Involvement of the PI3K/Akt/mTOR pathway in the enhancement of Glut1 by the IDH1/2 mutants.** (A) Phosphorylation of S6k and Akt in MEF-1WT, MEF-1MUT, MEF-2WT, MEF-2MUT, and the control cells. Expression of β-actin served as an internal control. (B) Involvement of Akts in the induction of Glut1. Akt1/2/3 were silenced with a mixture of *Akt1*, *Akt2* and *Akt3* (Akt1/2/3) siRNA for 48 h. (C and D) MEF-1MUT, MEF-2MUT and control cells were treated with 1 μM of PI-103 (C) or 20 nM of rapamycin (D) for 24 h. Lysate with/without the treatment were subjected to Western blotting.

presence of *IDH1/2* mutations. These results indicate that the PI3K/Akt/mTORC1-Hif1α axis is involved in the induction of Glut1 by the IDH1/2 mutants.

## Discussion

In this study, we have shown that oncogenic *IDH1/2* mutations induce the expression of Glut1 in MEF cells and HCT116 cells, and that activation of PI3K/Akt/mTOR pathway and up-regulation of Hif1α are involved in the induction of Glut1 (S3 Fig).

It is of note that the IDH2 mutant showed greater effects on the production of 2-HG *in vitro* compared with the IDH1 mutant and similar observations were shown in previous reports [19–21]. Since IDH1 and IDH2 is a cytosolic and mitochondrial enzyme, respectively,

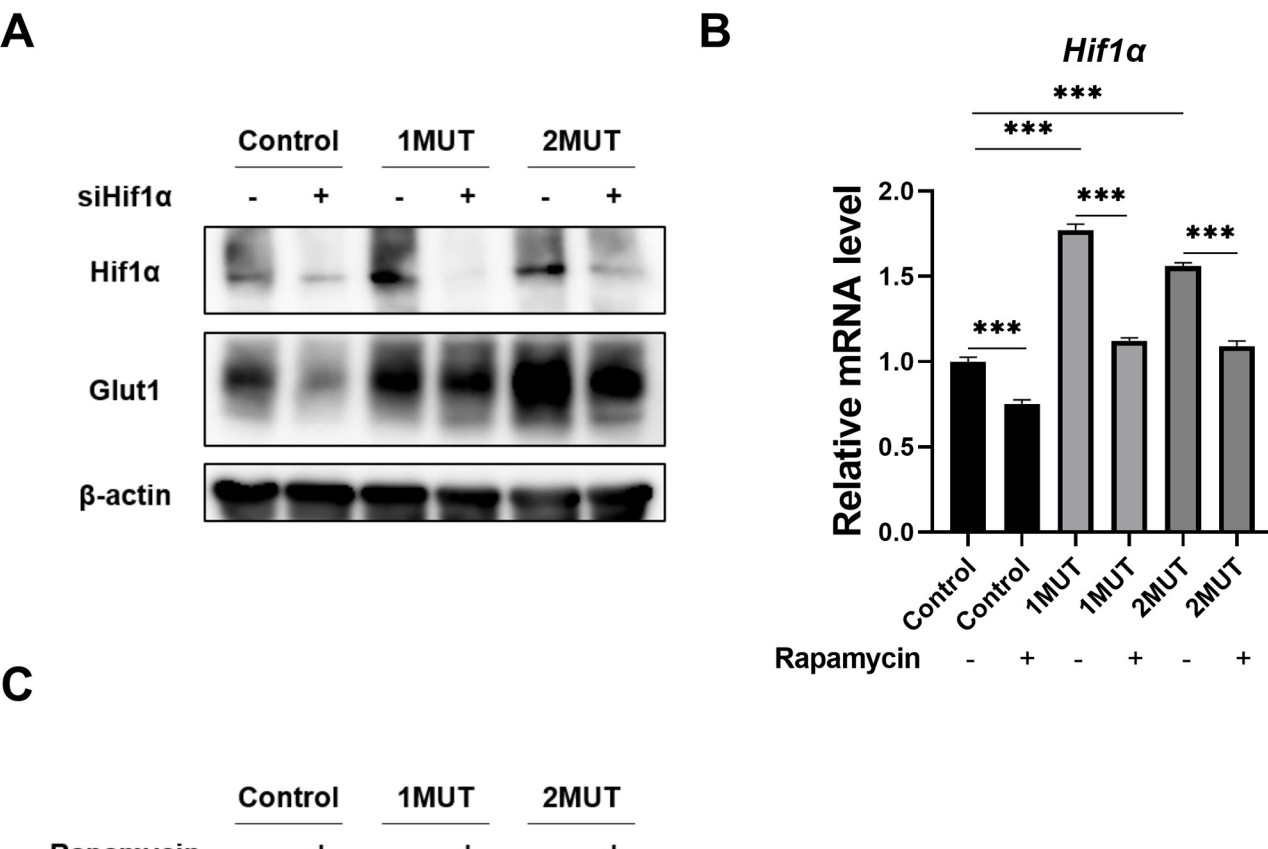

**Fig 6. Involvement of Hif1α in the induction of Glut1 in MEF cells.** (A) MEF-1MUT, MEF-2MUT, and the control cells were treated with *Hif1α* or control siRNA for 48 h. Expression of β-actin served as an internal control. (B) MEF-1MUT, MEF-2MUT, and the control cells were treated with 20 nM of rapamycin for 24 h. Expression was determined by real time-PCR. Expression of *Gapdh* was used as an internal control. The data represent mean ± SD of three independent experiments. *** p<0.001. (C) The lysates of MEF cells indicated in (B) were subjected to Western blotting.

IDH2 might have greater accessibility to α-KG than IDH1, resulting in a larger amount of 2-HG production. Consistent with this notion, it was reported that forced expression of mutant IDH1 that incorporated the N-terminal mitochondrial targeting sequence of IDH2 resulted in mitochondrial localization and a greater accumulation of 2-HG in cells [20].

Unexpectedly, both IDH1 and IDH2 mutants inhibited cell proliferation. *IDH1* and *IDH2* were generally considered as atypical oncogenes for their double-edged sword role in human carcinogenesis [22]. Although *IDH* mutations had been reported to promote cell proliferation in many types of cancer cells as a basal oncogenic activity [4], other studies also showed contradictory results of reduced cell growth induced by *IDH* mutations or 2-HG [23–25], suggesting that the effect of IDH mutants on cell proliferation is dependent on the type of cells. Interestingly, the GSEA analysis in the current study uncovered that of genes associated with the p53 pathway were enriched in MEF-2MUT cells (S2 Table), suggesting that the activated p53 signaling pathway may be involved in the suppressed proliferation of MEF-2MUT cells. In addition, previous research had revealed that 2-HG can competitively bind and inhibit ATP

synthase, resulting in decreased ATP contents, mitochondrial respiration and subsequent suppression of cell growth in *IDH1* mutant cells [26]. Wnt/β-catenin signaling was also identified to be involved in the inhibition of cell proliferation, migration and invasion in *IDH1* mutant glioblastoma cell lines [25]. These studies indicated that repressed cell proliferation by IDH1/2 mutants might be correlated with a complex mechanism regulated by cell energy metabolism or signaling pathways.

In this study, we clarified that Glut1 encoded by *Slc2a1* is a *bona fide* downstream target of the *IDH* mutations. In addition to Glut1, we found that cyclooxygenase-2 (COX-2) encoded by *Ptgs2* and laminin gamma-2 encoded by *Lamc2* are candidate molecules regulated by the *IDH* mutations through the accumulation of 2-HG. Considering the pleiotropic and multifaceted roles of COX-2 and laminin gamma-2 acting on tumorigenesis [27, 28], the regulation of these two molecules by the *IDH1/2* mutations should be investigated in future studies.

Previous research reported that mutant IDH1 activates glycolysis through the upregulation of PFKP in mouse intrahepatic biliary organoid [29]. On the other hand, 2-HG attenuates aerobic glycolysis in leukemia by targeting the FTO/m$^6$A/PFKP/LDHB axis [30]. These studies indicate that the effect of *IDH* mutation or 2-HG on glucose metabolism may depend on cell type. Our results showed that the *IDH1/2* mutations induced altered glucose metabolism, enhanced glucose uptake and glycolysis in MEF cells, which is similar to the Warburg effect acting in cancer cells [31]. Since mutant IDH1/2 enzymes convert α-KG into 2-HG, cancer cells carrying these mutants should have alterations in the concentration of TCA cycle intermediates, and require substitution of energy sources to attenuate the deficiency of metabolic substrates. It was suggested that lactate is actively imported and converted into α-KG in *IDH1* mutant gliomas, and that supplement of metabolic substrates is dependent on lactate, which can alleviate cells from the metabolic stress that results from defective isocitrate processing [32]. Therefore, increased Glut1 expression, glucose influx, and glycolysis by the *IDH1/2* mutations may function as a compensatory mechanism to rescue the aberrant aerobic respiration under the metabolic reprogramming. Although we have detected elevated glucose uptake in HCT116 cells, no alteration in intracellular lactate level was observed (S2D and S2E Fig). It has been reported that oncogenic *KRAS* mutation increases the expression of lactate dehydrogenase A and consequent production of lactate in HCT116 cells [33]. In addition, HCT116 cells grown in high-glucose condition have a higher lactate level than noncancerous fetal human colonocytes [34]. These findings may suggest that *KRAS* mutation and/or high-glucose condition weakened or canceled the effect of *IDH1/2* mutations on glycolysis through enhanced Warburg effect in HCT116 cells.

Elevated phosphorylation of S6k, Akt (Ser473) and Akt (Thr308) was detected in the MEF cells and HCT116 cells expressing IDH1/2 mutants, demonstrating the augmented activity of the PI3K/Akt/mTOR pathway. The underlying regulatory mechanism of Glut1 expression by *IDH* mutations-mediated PI3K/Akt/mTOR signaling was further investigated. Reduced expression of Glut1 by pharmacological and genetic inhibition of the PI3K/Akt/mTORC1 pathway was demonstrated in this study. Previous study has reported that trafficking of GLUT1 to the cell surface was mediated through Akt activation [35]. Additionally, Akt activation was suggested to be associated with the gene expression of Glut1 [36, 37]. Consistent with these reports, our results indicated that activation of the PI3K/Akt/mTORC1 cascade by the *IDH* mutations transcriptionally upregulates Glut1 expression, and suggested the involvement of transcription factor(s).

HIF1α is a transcription factor associated with metabolic alterations during tumorigenesis, and the protein is regulated through its degradation by prolyl hydroxylase domain-containing protein (PHD)-mediated hydroxylation and subsequent hydroxylation-targeted ubiquitination under normoxic condition [38]. It has been reported that reduced α-KG level by conversion to

2-HG might increase the level of HIF1α, as α-KG is necessary for PHD-mediated degradation of HIF1α [39, 40]. However, other lines of evidence showed that 2-HG stimulates the activity of the PHD, which results in the decreased expression of HIF1α [41]. Thus the mechanism(s) on how HIF1α was regulated in the context of *IDH* mutations remained controversial. In the present study, RNA-seq revealed that *Hif1α* level was increased in MEF-2MUT cells compared to the control cells, indicating transcriptional upregulation of *Hif1α* by the *IDH2* mutation. Western blotting additionally showed that exogenous expression of the IDH1 or IDH2 mutant increased the Hif1α protein, and that knockdown of Hif1α reduced Glut1 expression in MEF-1MUT and MEF-2MUT cells. These data corroborated the findings that Hif1α was also involved in the induction of Glut1 by the IDH1/2 mutants. Importantly, 2-HG treatment did not change the expression of *Hif1α* on mRNA level although Glut1 protein was induced by the treatment. This discrepancy may be explained by a report showing that Hif1α protein is stabilized by PHD inhibition in response to 2-HG [42]. In addition, previous studies demonstrated that mTOR activation regulates HIF1α expression by increased synthesis or stabilization of HIF1α [43, 44]. Our data confirmed that Hif1α expression is regulated by mTOR under the background of *IDH1/2* mutations.

In this study, Glut1 was identified as a downstream target molecule induced by the cancer-associated hotspot IDH1/2 mutants through a PI3K/Akt/mTORC1-Hif1α axis. Increased Glut1 expression consequently altered cellular glucose metabolism. This metabolic deregulation may partially contribute to development of human cancer carrying oncogenic *IDH1* and *IDH2* mutations. Future studies using normal epithelial cell lines or organoids and mouse models are required to further elucidate the roles of Glut1 in tumorigenesis associated with oncogenic *IDH1/2* mutations.

## Supporting information

**S1 Fig. IDH1 and IDH2 expression in MEF cells expressing cancer-associated *IDH1/2* mutations.** Western blot analysis of the MEF-1WT, MEF-1MUT, MEF-2WT, MEF-2MUT and control MEF cells. Expression of β-actin served as an internal control. The anti-IDH1 antibody recognized not only endogenously and exogenously expressed IDH1 but also exogenously expressed IDH2. The anti-IDH2 antibody recognized both endogenous and exogenous expression of IDH2. Expression of endogenous IDH2 is too low to be detected in the MEF cells.
(TIF)

**S2 Fig. Gene expression, 2-HG accumulation, cell proliferation and glucose metabolic analysis of HCT116 cells expressing cancer-associated *IDH1/2* mutations.** (A) Western blot analysis of the HCT116-1WT, HCT116-1MUT, HCT116-2WT, HCT116-2MUT and control HCT116 cells. Expression of β-actin served as an internal control. (B) Concentration of 2-HG in the lysates from cells indicated in (A). The data represent mean ± SD of three independent experiments. ** $p < 0.01$, *** $p < 0.001$. (C) The proliferation of cells was measured by WST-8 assay. The data represent mean ± SD of three independent experiments. *** $p < 0.001$. (D) Glucose uptake levels in HCT116-1WT, HCT116-1MUT, HCT116-2WT, HCT116-2MUT, and the control cells were measured by a bioluminescent assay based on the detection of 2DG6P. The data represent mean ± SD of three independent experiments. *** $p < 0.001$. (E) Intracellular lactate levels in the HCT116 cells indicated in (D) were measured by a bioluminescent assay for the detection of L-lactate. The data represent mean ± SD of three independent experiments.
(TIF)

**S3 Fig. Underlying mechanism of Glut1 expression regulated by mutant IDH1/2 through the PI3K/Akt/mTORC1-Hif1α axis.** Red solid arrows indicate the mechanism elucidated in this research. Black solid arrows show regulatory effects reported previously. Black dotted arrows exhibit potential regulations remaining unknown.
(TIF)

**S1 Table. List of the genes with altered expression by the IDH2 mutant or 2-HG treatment.**
(XLSX)

**S2 Table. Gene sets enriched in MEF-2MUT cells.**
(XLSX)

**S3 Table. Gene sets enriched in 2HG treated MEF.**
(XLSX)

**S1 Raw images.**
(PDF)

# Acknowledgments

We are grateful to Seira Hatakeyama, Yumiko Isobe, and Rika Koubo for their technical assistance.

# Author Contributions

**Conceptualization:** Xun Liu, Tsuneo Ikenoue.

**Formal analysis:** Xun Liu, Kiyoshi Yamaguchi, Kiyoko Takane.

**Investigation:** Xun Liu.

**Methodology:** Xun Liu, Chi Zhu.

**Project administration:** Yoichi Furukawa, Tsuneo Ikenoue.

**Resources:** Makoto Hirata, Yoko Hikiba, Shin Maeda.

**Software:** Xun Liu.

**Supervision:** Yoichi Furukawa, Tsuneo Ikenoue.

**Validation:** Xun Liu.

**Writing – original draft:** Xun Liu, Tsuneo Ikenoue.

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
