## [Decision Letter · Decision Letter 0]

6 May 2021

PONE-D-21-06211

Cancer-associated IDH mutations induce Glut1 expression and glucose metabolic disorders through a PI3K/Akt/mTORC1-Hif1α axis

PLOS ONE

Dear Dr. Ikenoue,

Thank you for submitting your manuscript to PLOS ONE. After careful consideration, we feel that it has merit but does not fully meet PLOS ONE’s publication criteria as it currently stands. Therefore, we invite you to submit a revised version of the manuscript that addresses the points raised during the review process.

The experiments in the manuscript are performed using human IDH1/2 overexpression in a single cell line, mouse embryonic fibroblasts. Using human/mouse metabolic chimeras as well as lack of confirmation in different system weakens authors conclusions. To generalize the findings, it is important to overexpress IDH1/2 in human cancer cell line that lacks their expression as well as knock down IDH1//2 in human cancer cell line that overexpresses these proteins to show the reverse phenotype such as decline in GLUT1 and Akt phosphorylation and upregulation of HIF1 signaling. The validation is especially important because authors observe a paradoxical phenotype: an inhibition of proliferation with the expression of cancer associated mutants of IDH1/2. In addition, even though authors provide an excel table with up and downregulated genes, the data should be deposited in the publicly available database. I would also suggest adding a schematic for the proposed changes in the activities of IDH1/2 mutants compared to the WT to the manuscript.

We look forward to receiving your revised manuscript.

Kind regards,

Irina U Agoulnik, Ph.D.

Academic Editor

PLOS ONE

Journal Requirements:

2) PLOS ONE now requires that authors provide the original uncropped and unadjusted images underlying all blot or gel results reported in a submission’s figures or Supporting Information files. This policy and the journal’s other requirements for blot/gel reporting and figure preparation are described in detail at https://journals.plos.org/plosone/s/figures#loc-blot-and-gel-reporting-requirements and https://journals.plos.org/plosone/s/figures#loc-preparing-figures-from-image-files. When you submit your revised manuscript, please ensure that your figures adhere fully to these guidelines and provide the original underlying images for all blot or gel data reported in your submission. See the following link for instructions on providing the original image data: https://journals.plos.org/plosone/s/figures#loc-original-images-for-blots-and-gels.

3) Please include captions for your Supporting Information files at the end of your manuscript, and update any in-text citations to match accordingly. Please see our Supporting Information guidelines for more information: http://journals.plos.org/plosone/s/supporting-information.

Reviewers' comments:

Reviewer's Responses to Questions

**Comments to the Author**

1. Is the manuscript technically sound, and do the data support the conclusions?

Reviewer #1: Partly

Reviewer #2: Yes

2. Has the statistical analysis been performed appropriately and rigorously? 

Reviewer #1: No

Reviewer #2: Yes

3. Have the authors made all data underlying the findings in their manuscript fully available?

Reviewer #1: No

Reviewer #2: Yes

4. Is the manuscript presented in an intelligible fashion and written in standard English?

Reviewer #1: Yes

Reviewer #2: Yes

5. Review Comments to the Author

Reviewer #1: 1.A critical defect of this study is that the authors used MEFs as their cell model. Since the authors wanted to establish a pathway that helps to explain the underlying carcinogenic mechanisms of IDH mutations, the results derived from MEFs might not be true in cancer or other types of cells.

2.Another critical defect is that the authors did not perform RNA-seq on the IDH1 mutation group, but tried to build a common regulatory mechanism for both IDH1 and IDH2. The rationale was wrong.

3.The authors showed altered glucose uptake and lactate production with IDH1/2 mutation in MEFs, which lacked verification in cancer cells or patient tissues with IDH1/2 mutations.

4.In Figure 1A, the authors should include western blotting results of IDH1 and IDH2 as well.

5.In Figure 1B, statistical analysis was lacking.

6.RNA-seq analysis details should be stated, e.g. the cutoff for DEG determination.

7.The authors said that “It is of note that rapamycin markedly decreased the Hif1α protein in these cells (Fig6C), suggesting that post-transcriptional regulation plays a major role in the expression of Hif1α.”. However, the RT-qPCR results in Figure 6B also showed that Hif1α mRNA was suppressed by rapamycin. Thus, there was no enough evidence to show that post-transcriptional regulation was involved.

8.The authors should provide results to show that IDH1/2 regulates Glut1 through Hif1α.

9.The raw RNA-seq data should be deposited to a public repository and the accession number should be included in the manuscript.

Reviewer #2: The authors examine the role of cancer-associated IDH mutations on the PI3K/Akt/mTORCH1-HIF-1a axis and subsequent Glut-1 expression and glucose metabolism. This is an interesting story but the manuscript might be improved with attention to the following comments and questions:

Methods: What was the rationale for using MEF cells? Why this model? What was the control retrovirus construct of the “control MEF” cells?

Results: How does “HIF-1 pathway” differ from “VEGF pathway”, isn’t VEGF at least partially if not fully controlled by HIF-1?

6. PLOS authors have the option to publish the peer review history of their article (what does this mean?). If published, this will include your full peer review and any attached files.

Reviewer #1: No

Reviewer #2: **Yes: **Randy Jensen

---

## [Author Response · Author response to Decision Letter 0]

28 Jul 2021

We sincerely thank the editor and reviewers for their thoughtful and positive comments to our manuscript. 

To editor:

As the editor suggested, to generalize the findings that we obtained using MEF cells, we generated HCT116 cells, a human colorectal cancer cell line, stably expressing mutant IDH1 and IDH2, and evaluated the 2-HG accumulation, cell proliferation, genes expression and metabolic characteristics in these cells as additional experiments.

We found that IDH1/2 mutations suppressed cell proliferation HCT116 cells as well as MEF cells. As we noted in the discussion section, although IDH mutations had been reported to promote cell proliferation in many types of cancer cells as a basal oncogenic activity, other studies also showed contradictory results of reduced cell growth induced by IDH mutations or 2-HG. Here, we demonstrated that IDH mutations suppress cell proliferation of HCT116 cells. 

IDH1/2 mutations induced accumulation of 2-HG, activation of PI3K/Akt/mTORC1-Hif1α axis, and enhancement of glucose uptake in HCT116 cells, all of which were consistent to the findings obtained in MEF cells, except the lactate production. We speculated that the enhanced Warburg effect in HCT116 cells reported in previous studies might weaken the effect of IDH mutants on glycolysis and made it difficult to detect the change of lactate levels induced by IDH1/2 mutations.

According to the editor’s suggestion, we have deposited the raw data to GEO database (GSE180369).

Furthermore, according to the editor’s constructive suggestion, we added a schematic of the underlying mechanism of Glut1 expression regulated by mutant IDH1/2 through the PI3K/Akt/mTORC1-Hif1α axis in supplementary Figure 3. 

To reviewer #1: 

1. We thank the reviewer’s comment. As the reviewer mentioned, we did not show the effect of exogenous expression of cancer-associated IDH1 and IDH2 mutations in cancer cells. Therefore, we generated HCT116 cells, a human colorectal cancer cell line, stably expressing mutant IDH1 and IDH2, and evaluated the 2-HG accumulation, cell proliferation, genes expression and metabolic characteristics in these cells as additional experiments.

2. We thank the reviewer’s constructive comment. Actually, we preserved the total RNA samples of MEF cells stably expressing mutant IDH1 (MEF-1MUT) when we performed RNA-Seq on the IDH2 mutation group (MEF-2MUT). Since our RNA-seq before aimed to clarify the effect of IDH mutant through accumulated 2-HG, we finally determined to use RNA sample of MEF-2MUT, whose 2-HG accumulation is significantly higher than MEF-1MUT. As a result, Slc2a1, encoding Glut1, was up-regulated in MEF-2MUT and 2-HG treated MEF cells. This time, we carried out RNA-seq in MEF-1MUT using the samples we kept before. Unfortunately, although we found increased read counts of Slc2a1 in MEF-1MUT cells compare to the control cells (Read counts of control cells: 227.718, 261.82, 290.418; read counts of MEF-1MUT: 342.557, 327.776, 484.593), there was no statistical significance between two groups. However, as shown in Figure 3C and 3G, both real-time PCR and western blotting results demonstrated that the Slc2a1 expression was up-regulated in both MEF-1MUT and MEF-2MUT. Therefore, a common regulatory mechanism for Slc2a1 should be induced by IDH1 and IDH2 mutants.

3. According to the reviewer’s comment, we analyzed glucose uptake and lactate production in HCT116 cells. As we expected, both mutant IDH1 and IDH2 significantly increased glucose uptake in HCT116 cells. However, lactate production seemed to be unaltered. Previous research reported that lactate production was augmented in HCT116 cells. This may suggest that the enhanced Warburg effect in HCT116 cells weakened the effect on glycolysis by IDH mutants and made it difficult to detect the change of lactate levels induced by IDH1/2 mutations. We mentioned this issue in the discussion section.

4. According to the reviewer’s comment, we added the western blotting results of IDH1 and IDH2 in supplementary Figure 1.

5. According to the reviewer’s comment, we changed the results in Figure 1B into new data of 2-HG measurement with statistical analysis. The data represent mean ± SD of three independent experiments.

6. According to the reviewer’s comment, we added the statement on the cutoff for DEG determination utilized in our RNA-seq analysis in the Materials and Methods section.

7. We thank the reviewer’s thoughtful suggestion. Real-time PCR results in Figure 6B showed that Hif1α mRNA was suppressed by rapamycin, whereas in Figure 6C, rapamycin markedly decreased the Hif1α protein expression. This difference between RNA and protein expression led us to the inference that post-transcriptional regulation may play a major role in the expression of Hif1α in context of IDH1/2 mutations. However, as the reviewer pointed out, we did not have any more evidence to support our speculation. Therefore, we agree that the sentence seems to be an overstatement and consider that both transcriptional and post-transcriptional may play a role in the expression of Hif1α in the presence of IDH1/2 mutations. We changed the description in the discussion section. 

8. In Figure 6A, we showed that Glut1 expression levels were reduced after Hif1α knockdown in MEF-1MUT and MEF-2MUT cells. We think these results suggest that IDH1/2 regulates Glut1 expression through Hif1α. 

9. According to the reviewer’s suggestion, we have already deposited our RNA-seq data to GEO database (GSE180369).

To reviewer #2: 

Comment on ‘Methods’

We thank the reviewer’s thoughtful comments on the methods of this study. Data in this paper actually is the in vitro part of a study aimed to establish and analysis of a novel mouse model of human cancer based on knockin of cancer-associated IDH1/2 mutations (Data not published). For the study of the functions of IDH1/2 mutations in tumorigenesis, it is desirable to use normal epithelial cells instead of cancer cells to exclude the influence of complex genetic and metabolic background in cancer cells. However, we did not have any available normal epithelial cells. Therefore, we chose MEF cells in this research. Organoids established from normal epithelial tissues can be utilized in future studies. The control MEF cells means MEF cells stably expressing pMX-control vector, officially named pMXs-puro vector.

Comment on ‘Results’

We thank the reviewer’s thoughtful comment of the results of this study. As the reviewer mentioned, VEGF pathway is a downstream pathway mainly regulated by HIF-1 pathway. Although genes involved in these two pathways are not necessarily common, Hif1α is one of the genes that are involved in both pathways. The GSEA based on our RNA-seq showed that genes associated with “HIF-1 pathway” were enriched in MEF cells expressing mutant IDH2 compared to the control cells, and the genes correlated with “VEGF pathway” were enriched by 2-HG treatment in MEF cells. These results indicated that Hif1α might be regulated by IDH mutation through 2-HG. Further, we proved our hypothesis that both transcriptional and post-transcriptional regulation play a role in the expression of Hif1α in the presence of IDH1/2 mutations using qRT-PCR and western blotting.

---

## [Decision Letter · Decision Letter 1]

24 Aug 2021

Cancer-associated IDH mutations induce Glut1 expression and glucose metabolic disorders through a PI3K/Akt/mTORC1-Hif1α axis

PONE-D-21-06211R1

Dear Dr. Ikenoue,

We’re pleased to inform you that your manuscript has been judged scientifically suitable for publication and will be formally accepted for publication once it meets all outstanding technical requirements.

Kind regards,

Irina U Agoulnik, Ph.D.

Academic Editor

PLOS ONE

Additional Editor Comments (optional):

Reviewers' comments:

Reviewer's Responses to Questions

**Comments to the Author**

1. If the authors have adequately addressed your comments raised in a previous round of review and you feel that this manuscript is now acceptable for publication, you may indicate that here to bypass the “Comments to the Author” section, enter your conflict of interest statement in the “Confidential to Editor” section, and submit your "Accept" recommendation.

Reviewer #1: (No Response)

Reviewer #2: All comments have been addressed

2. Is the manuscript technically sound, and do the data support the conclusions?

Reviewer #1: (No Response)

Reviewer #2: Yes

3. Has the statistical analysis been performed appropriately and rigorously? 

Reviewer #1: (No Response)

Reviewer #2: Yes

4. Have the authors made all data underlying the findings in their manuscript fully available?

Reviewer #1: (No Response)

Reviewer #2: Yes

5. Is the manuscript presented in an intelligible fashion and written in standard English?

Reviewer #1: (No Response)

Reviewer #2: Yes

6. Review Comments to the Author

Reviewer #1: (No Response)

Reviewer #2: (No Response)

7. PLOS authors have the option to publish the peer review history of their article (what does this mean?). If published, this will include your full peer review and any attached files.

Reviewer #1: **Yes: **Xiaopeng Shen

Reviewer #2: **Yes: **Randy L Jensen

---

## [Editor Report · Acceptance letter]

3 Sep 2021

PONE-D-21-06211R1 

Cancer-associated IDH mutations induce Glut1 expression and glucose metabolic disorders through a PI3K/Akt/mTORC1-Hif1α axis 

Dear Dr. Ikenoue:

I'm pleased to inform you that your manuscript has been deemed suitable for publication in PLOS ONE. Congratulations! Your manuscript is now with our production department. 

Kind regards, 

on behalf of

Dr. Irina U Agoulnik 

Academic Editor

PLOS ONE